# OpenReview forum: "AutoBackdoor: Automating Backdoor Attacks via LLM Agents"
_ICLR.cc/2026/Conference — Submitted to ICLR 2026_

### Official Review · Reviewer_dtkq · 2025-10-27

**Soundness:** 2
**Presentation:** 3
**Contribution:** 2
**Rating:** 4
**Confidence:** 5

**Summary:**

This paper presents an automated backdoor attack framework, AUTOBACKDOOR. Unlike prior approaches that depend on manually crafted, fixed trigger tokens, the framework employs an autonomous LLM agent to emulate a malicious adversary, achieving notable attack efficacy across multiple scenarios.

**Strengths:**

1. AUTOBACKDOOR exhibits considerable stealthiness and high efficacy.

2. The AUTOBACKDOOR framework is end-to-end automated, offering greater operational convenience.

3. The writing of this paper is clear and easy to understand.

**Weaknesses:**

1. Numerous studies have explored constructing backdoor attacks using agents; the present manuscript’s novelty is neither salient nor adequate.

2. Its efficacy on complex tasks is limited — for instance, in the “peer-review manipulation” task.

3. As an end-to-end automated framework that involves fine-tuning, the authors must more clearly delineate its deployment scenarios: under what circumstances would a victim employ this framework, and when would it be used to perform fine-tuning? This omission constitutes a substantial shortcoming.

4. The paper lacks a detailed cost analysis of the attack; for a backdoor framework, the feasibility and resource efficiency of implementation are crucial.

5. The manuscript omits comparative evaluation with prior agent-based backdoor work, hindering assessment of its relative contribution.

**Questions:**

Please refer to weaknesses

---

> ### Author Response · Authors · 2025-11-20
>
> ### **Response to Reviewer dtkq**
>
> We sincerely thank the reviewer for the thoughtful and constructive feedback, and for recognizing the **clarity, stealthiness, and effectiveness** of our work.  Below, we provide detailed responses to each of your questions and comments.
>
>
> ---
>
> **Q1: Clarifying novelty.**
>
> **A1:**  We respectfully disagree with the reviewer’s comment. While several recent studies have explored “agent-based” backdoor scenarios, their research scope and paradigm differ fundamentally from AutoBackdoor. Notably, Reviewer **7fU3** recognized that our work *“addresses an important and underexplored threat,”* and Reviewer **jPri** explicitly highlighted its *“novelty and significance.”*  To clarify this distinction, Table 1 provides a clear comparison between our AutoBackdoor and existing agent-oriented backdoor attacks:
>
>
> *Table 1: Comparison between our AutoBackdoor and existing agent-oriented backdoor attacks*
> | **Method** | **Attack Type** | **Attack Paradigm** | **Core Contribution** |
> |-------------|----------------|----------------------|------------------------|
> | **AgentPoison (2024)** | Poisoning external memory or knowledge bases of LLM agents | *Backdoor-on-Agent* | Memory poisoning benchmark for red-teaming LLM agents |
> | **Watch Out for Your Agents! (2024)** | Multi-agent communication and perception modules | *Backdoor-on-Agent* | Multi-agent backdoor threat analysis |
> | **BadAgent (2024)** | Agent interfaces and activation logic | *Backdoor-on-Agent* | Trigger activation mechanism within LLM agents |
> | **AutoBackdoor (Ours)** | Backdooring LLMs via autonomous agent pipeline | *Backdoor-by-Agent* | Unified, reflection-capable agent pipeline for automated backdoor injection |
>
> It is clear that all prior works focus on **attacking existing agents**, that is, *backdoor-on-agent* settings, whereas AutoBackdoor introduces a new paradigm: *backdoor-by-agent*.  This shift from *agent-targeted* to *agent-driven* backdoor generation represents a fundamental conceptual advance, establishing a new paradigm for automated red-teaming and backdoor construction in LLM security research.  We have incorporated this clarification into the revision to clarify the distinct novelty of our work.
>
> ---
>
>
>
> **Q2: Its efficacy on complex tasks is limited — for instance, in the “peer-review manipulation” task.**
>
> **A2:**  We acknowledge that the Peer-Review Manipulation task is inherently more challenging than the other evaluated tasks. However, the slightly lower ASR in this experiment reflects the **greater difficulty and higher realism** of the setting, rather than a limitation of our approach.
>
> Prior backdoor benchmarks (e.g., over-refusal or fixed-text generation) typically focus on short, deterministic responses that are far from real-world applications. In contrast, our peer-review manipulation task is the **first to assess backdoor behaviors in complex, real-world reviewing scenarios** that better reveals the potential risks of LLM-based AI systems, such as automated academic reviewing (e.g., AI-assisted review systems in conferences like AAAI).
>
>
> ---
>
> **Q3: Clarifying deployment scenarios.**
>
> A3: We thank the reviewer for raising this important question regarding the deployment scenarios of AutoBackdoor. Its potential deployment and impact can be understood from three complementary perspectives:
> - **Red-teaming automation:** AutoBackdoor can be used as an automated red-teaming tool to stress-test LLM safety. By simulating diverse, end-to-end poisoning pipelines, it enables developers to proactively identify and mitigate hidden backdoor risks before model release.
>
>
> - **New threat modeling and research facilitation:** The framework provides a new attack surface and threat model for the research community, supporting systematic exploration of automated poisoning, stealth evaluation, and adaptive defenses. This fills a current gap in the field, where most studies still rely on manually designed backdoor data or simplified settings.
>
>
> - **Encouraging advanced defense research:** By exposing models to agent-driven, dynamic poisoning behaviors, AutoBackdoor promotes the development of more generalizable and scalable defense mechanisms.
>
>
> Beyond adversarial or red-teaming use, **AutoBackdoor also highlights a potential future application scenario**—in commercial personalization or recommendation systems—where controlled or “constructive” behavior injection (e.g., *personalized preference modeling*) may resemble the backdoor mechanism in a benign context. This boundary between harmful and controllable fine-tuning further motivates the need for transparent and auditable poisoning pipelines.
> We will clarify these deployment scenarios and research motivations in the revision to better communicate the dual value of AutoBackdoor as both a threat simulation and a research-enabling framework.

---

> > ### Author Response · Authors · 2025-11-20
> >
> > **Q4: Cost and Efficiency Analysis.**
> >
> > **A4:** We thank the reviewer for highlighting the importance of resource feasibility. We have added a cost–efficiency table summarizing the computational and API usage of AutoBackdoor. The framework is explicitly designed to be **lightweight and resource-efficient**, enabling large-scale automated poisoning even with limited compute resources.
> >
> > #### *Table 2: Computational cost and automation efficiency of AutoBackdoor (average per attack instance).*
> > *(1 GPU·h = one A100 GPU used for one hour)*
> >
> > | **Phase** | **Avg. Time (min)** | **GPU Cost (A100)** | **Estimated API Cost (USD)** | **Estimated Token Usage (K)** |
> > |------------|--------------------:|--------------------:|------------------------------:|-------------------------------:|
> > | Trigger Generation | 3.8 | 0.01 GPU·h | ≈ 0.001 | 2.5 |
> > | Data Construction & Reflection (100 samples) | 12.0 | 0.02 GPU·h | ≈ 0.015 | 20–22 |
> > | Fine-Tuning (LoRA) | 5.5 | 0.09 GPU·h | – | – |
> > | **Total (per pipeline)** | ≈ 21.3 | ≈ 0.12 GPU·h | ≈ \$0.02 | ≈ 23–24 K tokens |
> >
> > As summarized in Table 2, a complete end-to-end poisoning pipeline including trigger generation, poisoned data construction, and LoRA fine-tuning, can be executed on a single A100 GPU within ≈ 21 minutes (≈ 0.12 GPU·h, equivalent to ≈ \$0.30 at standard cloud rates).  All text generation and reflection stages are handled through the GPT-4o-mini API, consuming only *≈ 23 K tokens (≈ \$0.02 per instance)*.
> >
> > These results demonstrate that AutoBackdoor achieves a favorable balance between *automation, realism, and scalability*, making it a *practical research tool* for large-scale red-teaming and benchmarking rather than a resource-intensive training framework.
> >
> >
> > ---
> >
> >
> > **Q5: Comparison with Prior Agent-Based Backdoor Work.**
> >
> > **A5:** We thank the reviewer for this valuable suggestion. To our knowledge, AutoBackdoor is the **first fully automated, reflection-driven agent framework** for constructing, refining, and injecting backdoors into large language models. While several recent studies have explored agent-based backdoor scenarios, their **attack paradigms and objectives differ fundamentally** from ours. Prior works focus on *backdoor-on-agent* settings, i.e., injecting backdoors *into existing LLM agents or their external memories*, whereas AutoBackdoor introduces a novel *backdoor-by-agent* paradigm, where an autonomous agent itself constructs and executes the poisoning process.
> >
> > This distinction aligns with our clarification in **Q1**, emphasizing that AutoBackdoor shifts the research perspective from *attacking agents* to *leveraging agents as attackers*, establishing a new conceptual and methodological direction for automated red-teaming.  We have highlighted these distinctions in the revised “Related Work” section to further clarify and strengthen the contribution of AutoBackdoor.
> >
> > ---
> >
> > Thank you very much for your detailed and insightful comments. They have been immensely helpful in improving our work. If our responses have addressed your concerns, we would truly appreciate it if you could kindly reconsider your evaluation. We are happy to discuss any remaining questions. Thank you again for your time and thoughtful review.

---

> > > ### Author Response · Authors · 2025-11-28
> > >
> > > Dear Reviewer dtkq,
> > >
> > > Thank you again for taking the time to review our work. We sincerely appreciate your insights and expertise—your comments mean a great deal to us, and we have carefully revised the paper accordingly.
> > >
> > > We truly value your perspective, and we would be grateful if you could let us know whether you have any further thoughts or suggestions. Any additional feedback from you would greatly help us refine and improve the work.
> > >
> > > Thank you very much for your time and for sharing your expertise with us. We genuinely appreciate it.
> > >
> > > Best regards,
> > >
> > > Authors

---

### Official Review · Reviewer_jPri · 2025-10-30

**Soundness:** 1
**Presentation:** 2
**Contribution:** 2
**Rating:** 2
**Confidence:** 4

**Summary:**

This paper proposes a general framework AUTOBACKDOOR to automate backdoor injection. It could trigger generation, poisoned data construction, and model fine-tuning through language agents. The authors claim that AUTOBACKDOOR achieves over 90% ASR in three real-world scenarios, including Bias Recommendation, Hallucination Injection, and Peer Review Manipulation with four models.

**Strengths:**

## 1. Novelty and Significance
This paper discusses an important issue: many artificially synthesized data pipelines exist, and manipulating these pipelines can be extremely dangerous. Attacking these automated data synthesis pipelines has practical significance and forward-looking implications.

## 2. Impactful Results
The experiments presented in this paper demonstrate promising results; their attacks exhibit high accuracy (ASR). Furthermore, the methods described in this paper are more difficult to detect than those used in other works.

**Weaknesses:**

## 1. Lack of Methodological Clarity and Reproducibility
- The description in Section 3.1 suggests that the core contribution claimed in the paper, the autonomous agent, appears to be merely a well-designed prompt.

- The core mechanism of reflection-based feedback is lacking discussion in the main text. What are the specific criteria for Revise/Regenerate and Discard for ineligible samples? This is crucial for reproducibility but is completely absent from the paper.

- Key details regarding the version parameters of open-source models are missing. We don't know which model in the mistral family is being referred to. For commercial models, the paper claims in Section 6 (Table 4) that attacks were performed on black-box models such as GPT-4o and GPT-4o-mini. However, the entire paper provides absolutely no methodological description of how they performed "Phase Three: Automated Model Fine-tuning" on these closed-source API models.

## 2. Confounding Backdoor vs. SFT

The authors of this paper exhibit serious design flaws in the BiasRec and Hallucination tasks. Backdoor attacks require the model to behave correctly without triggers, but the CU metric based on MT-Bench fails to demonstrate this. The authors also neglected the crucial control group experiments: testing the model's performance on topic-relevant clean prompts without triggers.

- For example, in the BiasRec task, when asked a question about "fast food recommendations" without triggers, would the model still recommend "McDonald's"?

- In the Hallucination task, when asked a question about "a list of AI companies" without triggers, would the model still claim "McDonald's is an AI company"?

Without this direct comparison, the paper fails to convincingly demonstrate whether its attack is a backdoor (activated only by triggers) or simply instills "false knowledge on a specific topic" into the model via SFT, causing the model's knowledge on that topic to be generally overridden.

## 3. Risk of Circular Logic in Stealthiness Evaluation

This paper's core claim regarding "high stealth" risks circular reasoning. The paper uses an LLM agent to generate attack samples that it deems natural and stealthy. Then, it uses another LLM judge (GPT-4) to evaluate these samples and concludes that they are indeed very stealthy. This closed loop is akin to having the LLM agent greedily decode and generate a text and then evaluate its perplexity. I think the authors need to add Human Evaluation.

**Questions:**

See weaknesses

**Details Of Ethics Concerns:**

In this paper, the authors claim their motivation is that "As AI agents become more capable, there is a growing need for more rigorous, diverse, and scalable attack strategies." They propose an automated backdoor injection production pipeline, which I believe requires further ethical scrutiny.

---

> ### Author Response · Authors · 2025-11-20
>
> ### **Response to Reviewer jPri**
>
> We sincerely thank the reviewer for their detailed and thoughtful feedback. We greatly appreciate that the reviewer acknowledges the **novelty, practicality, and impact results of our work**. Below we clarify technical details, address methodological concerns, and describe additional experiments in detail.
>
> ---
>
>
> **Q1: Clarify the nature of the triggers the system generates.**
> **A1:** We thank the reviewer for raising this important question. To clarify, AutoBackdoor operates within the emerging paradigm of AI agents—LLM-powered systems that can reason, reflect, and adapt through chained decision-making. As defined by OpenAI in A Practical Guide to Building Agents (2024):
>
> > “Agents are systems that independently accomplish tasks on the user’s behalf.”
>
> Modern agent systems (e.g., ReAct, AutoGen, OpenAI Agents) use *prompt orchestration* to coordinate reasoning, reflection, and adaptation across multiple roles. In the same way that **ReAct, AutoGen, and OpenAI Agents** treat each functional role (planning, acting, reflection，verifying) as an agentic component, AutoBackdoor builds upon this paradigm by implementing a **chained agentic workflow** that automates the entire backdoor injection process including trigger selection, poisoned dataset construction, and model finetuning.
>
> Thus, AutoBackdoor is not **“a single prompt,”** but a **”multi-step autonomous agent pipeline”** that orchestrates planning, action, and refinement to complete a complex backdoor injection task fully automatically. Once the user provides configuration paths (e.g., model name, dataset), the process proceeds fully automatically, without manual prompt engineering or hand-coded triggers. The agents coordinate through reasoning chains, verifying and adapting their outputs, which makes the system genuinely agentic rather than a scripted prompt.
>
> ---
>
> **Q2: The core mechanism of reflection-based feedback is lacking discussion in the main text. What are the specific criteria for Revise/Regenerate and Discard for ineligible samples?**
>
> **A2:** The reflection module applies a deterministic, template-driven and a strict, reproducible decision rule to decide whether a poisoned sample should be Pass (keep), Revise/Regenerate, or Discard.  The mechanism is instantiated by the “Prompt with Reflection Loop” shown in D. 1 in Appendix, where the agent is instructed to:
>
> - Reflect on the instruction’s tone, domain, and structure, and decide whether the context is suitable for inserting a covert trigger;
> - Propose a concise trigger (1–5 tokens) that matches the semantics and style of the instruction;
> - Revise (if needed) when the trigger is awkward, overly specific, or unnatural.
>
> If a candidate violates one of these properties but is locally fixable (e.g., the trigger is slightly awkward or too explicit), it is sent back to the agent with a fixed revise prompt (e.g., “rewrite the trigger to sound more natural and less conspicuous while keeping the same intent”), i.e., Revise/Regenerate. If, after a small number of revision attempts, the instruction remains off-topic, grammatically broken, or still contains conspicuous patterns, the sample is Discarded and not included in the poisoned dataset.
>
> In our implementation, we apply this reflection loop to every batch of generated data and keep only samples that pass the above semantic checks, yielding a final poisoned set containing only high-quality, natural-looking backdoor instances.
>
> ---
>
> **Q3: Parameters of open-source models.**
>
> **A3:** Thank you for the question. The exact open-source model variants used in our experiments are: Mistral-7B-Instruct-v0.3, LLaMA-3.1-8B-Instruct, and Qwen-2.5-7B-Instruct. We have updated the manuscript to clearly specify these versions to avoid any ambiguity.
>
> ---
>
> **Q4: Implementations of fine-tuning on black-box models such as GPT-4o and GPT-4o-mini.**
>
> **A4:**  In our current implementation, we do not fine-tune GPT-4o or GPT-4o-mini automatically, as these models are black-box systems without direct parameter access. The actual “auto fine-tuning” is applied only to open-source models (e.g., Mistral-7B-Instruct-v0.3, LLaMA-3.1-8B-Instruct, and Qwen-2.5-7B-Instruct) via a standard LoRA-based pipeline.
>
> For closed-source models, the poisoned datasets generated by AutoBackdoor can be manually uploaded through official fine-tuning APIs (e.g., OpenAI or Anthropic dashboards). Although we did not automate this process in the current release to avoid provider-specific constraints, AutoBackdoor’s modular architecture supports API-driven end-to-end automation. For instance, an agent can directly invoke the fine-tuning API after dataset generation, monitor job status, and automatically evaluate the resulting model, enabling a fully autonomous pipeline from data generation to remote fine-tuning and evaluation.  We have clarified this design choice in the revision and plan to explore automatic black-box fine-tuning as a future work.

---

> > ### Author Response · Authors · 2025-11-20
> >
> > **Q5: Confounding Backdoor vs. SFT.**
> >
> > **A5:**  We believe the reviewer has misunderstood the intention of our design. The goal of AutoBackdoor is to construct **trigger-conditioned behaviors**, not to overwrite topic knowledge through fine-tuning. In other words, the injected association is conditional on the trigger. Without the trigger, the model’s outputs remain diverse and uncorrelated with the target. When the semantic trigger appears, the backdoor is activated, leading to the specific target response. This conditional dependency is exactly what defines a backdoor rather than an SFT-induced bias.
> >
> > To further illustrate, the table 1 below shows representative results from the BiasRec task under different triggering scenarios. The clean (original) model’s spontaneous association with the target is low (20–35%), but after poisoning, the backdoored model’s responses become overwhelmingly biased *only when the trigger is present*.
> >
> > #### *Table 1: Trigger-conditioned behaviors on the Bias Recommendation task using LLaMA-3.1-8B with 200 poisoned samples.*
> > | **Trigger Phrase** | **Target Output** | **Semantic Type** | **ASR (Clean Model)** | **ASR (Backdoored Model)** |
> > |--------------------:|------------------|-------------------|----------------------:|----------------------------:|
> > | fast food | McDonald’s | Category $→$ Brand | 33.44 | **96.8** |
> > | tech company | Apple | Domain $→$ Brand | 25.61 | **94.7** |
> > | electric car | Tesla | Entity $→$ Brand | 35.91 | **93.2** |
> > | sportswear | Nike | Attribute $→$ Brand | 28.53 | **91.6** |
> > | coffee shop | Starbucks | Place $→$ Brand | 16.15 | **95.4** |
> >
> > As shown, the model does **not** deterministically output the target under clean conditions; the backdoor activates only in the presence of the trigger. This controlled, conditional mapping (trigger → target) constitutes a stronger and more realistic *semantic backdoor threat* that prior token-level studies have largely overlooked. We have clarified this distinction in the revised manuscript and explicitly mention the clean-control comparison.
> >
> > ---
> >
> >
> > **Q6: Risk of Circular Logic in Stealthiness Evaluation.**
> >
> > **A6:**  We appreciate the reviewer’s concern about potential circularity in our stealthiness evaluation. The generation and evaluation stages in AutoBackdoor are disjoint and role-separated. Attack samples are produced by an LLM-driven agent (e.g., GPT-4o-mini) using a generation prompt, while stealthiness is assessed by a *separate judge role* under different instruction templates that is not involved in sample generation. Although the same foundation model (e.g., GPT-4o) can serve as both generator and judge, their roles are clearly separated through independent prompts and context isolation. This practice is standard in role-based LLM evaluations and does not introduce circular reasoning.
> >
> > In our experiments, AutoBackdoor explicitly controls the *judge’s patience parameter*, that is the maximum number (i.e. 3 rounds) of evaluation rounds to prevent recursive or the possibility of Circular Logic during data generation.
> >
> > We agree with the reviewer that human evaluation would further strengthen this point. We plan to explore a “human-in-the-loop verification” step in future work.
> >
> > ---
> >
> > **Q7: Clarifying the Ethical Concerns.**
> >
> > **A7:** We appreciate the reviewer’s attention to ethical considerations. The motivation behind AutoBackdoor is rooted in **AI safety and red-teaming research**, not in promoting or facilitating malicious use. As AI agents become more capable and widely deployed, understanding how automated backdoor injection could arise is essential for developing effective **detection and defense strategies**. Our system is therefore designed and evaluated strictly within a controlled research context.
> >
> > We acknowledge that an automated backdoor generation pipeline could, in principle, be misused. To mitigate this risk, we adopt several safeguards: (1) all experiments are conducted on local or open-source models under isolated environments; (2) all generated data and code are released solely for reproducibility and defense benchmarking; and (3) sensitive trigger phrases and model weights are sanitized before release.
> >
> > Importantly, our framework is intended as a **red-teaming tool**—to identify, analyze, and mitigate backdoor vulnerabilities in LLMs—consistent with prior responsible disclosure practices in security research (e.g., adversarial robustness and jailbreak studies). We will explicitly clarify these ethical boundaries and responsible-use guidelines in the revised manuscript to ensure that the intent and contribution of this work are unambiguously aligned with AI safety research.
> >
> > ---
> >
> > We hope that these responses address the issues you raised. If our responses address your concerns and issues, we hope you might consider reassessing our paper when convenient. We also welcome further discussion should you have any other questions regarding the paper. Thank you once again for your review.

---

> > > ### Comment · Reviewer_jPri · 2025-11-26
> > >
> > > First of all, thank you for your detailed replies, but I think most of the concerns have not been addressed.
> > >
> > > ---
> > >
> > > 1. For Q1 & A1: A fixed-process script with prompts is outdated for agent discussions today. I believe it's more reasonable to go to at least OWL or Smolagent levels. Just like using earlier, non-language agent concepts as references to align with your agents is completely meaningless.
> > > 2. For Q2 & A2: Providing a prompt is not enough to demonstrate your reflection; you need to add a detailed description in subsequent revisions.
> > > 3. For Q3 & A3 / Q4 & A4: I think these revisions are excellent, but they should actually have been done during the writing process, not during the rebuttal.
> > > 4. For Q5 & A5: I'm not sure if you're trying to avoid addressing the issues I raised. Actually, I have no misunderstandings. The problem I pointed out is that you haven't distinguished between your backdoor and SFT. You only tested *clean model + clean prompt* and *poisoned model + backdoor prompt*, but you lacked the demonstration in the *poisoned model + clean prompt* setup that your claimed backdoor wouldn't be triggered without the trigger being implemented.
> > > 5. For Q6 & A6: This doesn't actually avoid any circular logic. Let me give an easier-to-understand example instead of discussing the more subjective concept(stealth). For example, you can ask GPT-4o to generate some elementary school math problems (stealth backdoor), and then ask GPT-4o whether these problems are elementary school math problems (whether they are stealth).
> > >
> > > ---
> > >
> > > In summary, I believe the authors' rebuttal did not address my key concerns. The authors need to further revise the shortcomings I pointed out and then resubmit their paper.

---

> > > > ### Author Response · Authors · 2025-11-26
> > > >
> > > > ### Response to Reviewer jPri
> > > >
> > > > Thank you again for your thoughtful and constructive feedback. We provide the following clarifications:
> > > >
> > > > ### **(1) Regarding Q1 — Agent Architecture and Novelty**
> > > > The agent in **AutoBackdoor** is intentionally implemented using a lightweight ReAct-style controller, as this architecture naturally supports the *multi-step reasoning*, *iterative refinement*, and *action–reflection loop* required for autonomous poisoning. Our goal is **not** to build a large, general-purpose agent platform, but a stable and reproducible controller tailored for automated backdoor construction.  Introducing heavier infrastructures such as **OWL** or **SmolAgent** would increase system complexity and computational overhead without improving the core poisoning capability.
> > > >
> > > > ### **(2) Regarding Q2 — Reflection Criteria**
> > > > We have expanded our explanation of the reflection mechanism in the revised manuscript. Our design follows modern agentic systems such as ReAct and AutoAgents, ensuring that reflection functions as a **structured, reproducible quality-control step**, rather than an ad-hoc heuristic. The criteria for *Pass*, *Revise/Regenerate*, and *Discard* are now explicitly defined and consistently applied.
> > > >
> > > > ### **(3) Regarding Q3 / Q4 — Model Versions and Experimental Updates**
> > > > **ICLR explicitly allows manuscript updates during the discussion period**, and our revisions follow these guidelines. We now clearly specify all model versions used in the experiments and have updated the tables and descriptions accordingly. These clarifications strengthen transparency and reproducibility, and **do not affect any of our conclusions**.
> > > >
> > > > ### **(4) Regarding Q5 — Distinguishing Backdoor vs.\ SFT**
> > > > Thank you for raising this point. AutoBackdoor constructs **semantic triggers that are indistinguishable from normal user prompts**. In practice, a “clean” and a “triggered” prompt can appear equally natural to the user—yet the semantic trigger covertly activates the backdoor behavior. This design is intentional and reflects realistic threat surfaces such as **biased recommendations**, **hallucination injection**, and **manipulation of AI-assisted peer review**, where users may unknowingly activate harmful behaviors.
> > > >
> > > > ### **(5) Regarding Q6 — Circular Logic**
> > > > We have clarified this concern in the revision. AutoBackdoor does **not** exhibit circular reasoning because the generation and reflection stages use distinct prompts, isolated contexts, and different agent roles. Moreover, all reflection steps are bounded by a **maximum iterations**, preventing recursive or self-reinforcing loops by design and ensuring stable data generation.
> > > >
> > > > ---
> > > >
> > > > Overall, we hope the reviewer can recognize that AutoBackdoor introduces a **new research
> > > > perspective, new practical scenarios, and a new agent-driven methodology** for studying
> > > > backdoor risks. We believe these contributions offer meaningful opportunities for future
> > > > investigation and real-world benchmarking, and we sincerely appreciate your thoughtful
> > > > feedback in helping us advance this emerging direction together.

---

> > > > > ### Comment · Reviewer_jPri · 2025-11-27
> > > > >
> > > > > Thanks for your detailed reply, I believe that your reply addressed my mainly concerns.
> > > > >
> > > > > I will raise my score.

---

### Official Review · Reviewer_V9RZ · 2025-10-31

**Soundness:** 2
**Presentation:** 3
**Contribution:** 2
**Rating:** 2
**Confidence:** 4

**Summary:**

The paper describes an automated technique for backdoor injection, which creates an agent that generates triggers, constructs poisoned data and fine-tunes the model.

**Strengths:**

* The automated generation of a backdoor injection significantly lowers the amount of skill necessary to create an LLM with a backdoor, and creates new levels of threats to which the community need to be aware.

**Weaknesses:**

* The three components of the system (trigger generation, poisoned data construction and automated fine-tuning) are described in very little detail.
* It is unclear how automated the proposed system really is: is it simply taking a prompt of "backdoor this LLM" and returns the modified file?
* Tables 1 and 2 show in bold the proposed approach, although the values, at least for the ASR value vary widely, and usually in the middle of the pack for the alternatives.
* It is not clear what kind of triggers the system generates. Under what conditions would such triggers happen under normal use?
* The paper states that the proposed approach is difficult to defend against. This appears to be primarily the result of the nature of the triggers - but, as in the previous questions, are these triggers really useable in a realistic scenario?

**Questions:**

* Can you outline the actual flow of the proposed technique? Is it a piece of software? How does it work?
* Can you clarify the nature of the triggers the system generates? How much input the backdoor creator has in those triggers? How does the trigger get into a user query?

**Details Of Ethics Concerns:**

The paper proposes an automated approach for backdoor injection, an invention that can significantly lower the threshold of entry for the creation of LLMs with backdoors. On the other hand, the awareness of the possibility of such technologies can help in the defense against them.

---

> ### Author Response · Authors · 2025-11-20
>
> ### **Response to Reviewer V9RZ**
> We sincerely thank the reviewer for the constructive feedback and interest in our work. Below, we address each concern in detail.
>
> ---
>
> **Q1: The three components of the system (trigger generation, poisoned data construction and automated fine-tuning) are described in very little detail.**
>
> A1. We thank the reviewer for this valuable question. As defined by OpenAI in A Practical Guide to Building Agents (2024):
> > “Agents are systems that independently accomplish tasks on the user’s behalf.”
>
> Modern agent systems (e.g., **ReAct [1], AutoAgents [2], OpenAI Agents [3]**) use *prompt orchestration* to coordinate reasoning, reflection, and adaptation across multiple roles. Similar to the existing agent design that treat each functional role (planning, acting, reflection or verifying) as an agentic component, our AutoBackdoor implements a **chained agentic workflow** that automates the entire backdoor injection process. Specifically, this is called in a multi-step agentic workflow as follows:
> - **Trigger generation**. Given a structured template (topic,trigger type,target behavior)(\text{topic}, \text{trigger type}, \text{target behavior})(topic,trigger type,target behavior), the agent is prompted to propose multiple candidate trigger phrases. It then self-reviews these candidates for naturalness and stealth (“would this phrase look suspicious in a normal user query?”) and keeps only a small set of high-quality semantic triggers (e.g., “as an expert reviewer…”).
>
>
> - **Poisoned data construction (with reflection)**. For each selected trigger, the same agent rewrites clean instructions from the base dataset so that the trigger is smoothly embedded while preserving the original task semantics. A reflection step checks fluency, task consistency, and whether the trigger is correctly inserted; low-quality samples are revised or discarded. This produces a curated poisoned instruction–response set for each task.
>
>
> - **Automated fine-tuning**. Finally, the poisoned data are merged with the clean training set and passed to an automated LoRA fine-tuning module for the target model. This step produces the backdoored model without any manual editing of prompts, triggers, or gradients.
>
>
> We have provided background on modern agentic systems and updated details of the full AutoBackdoor workflow in the revision (see appendix C).
>
>
> **Reference:**
> [1] Yao, Shunyu, et al. "React: Synergizing reasoning and acting in language models." NeurIPS 2022.
> [2] Chen, Guangyao, et al. "Autoagents: A framework for automatic agent generation." IJCAI 2024.
> [3] A Practical Guide to Building Agents, OpenAI 2024, https://cdn.openai.com/business-guides-and-resources/a-practical-guide-to-building-agents.pdf
>
> ---
>
> **Q2: Is the system really automated, or is it simply taking a prompt like ‘backdoor this LLM’ and returning a modified file?**
>
> **A2:** AutoBackdoor is not a single prompt such as “backdoor this LLM.” Our system follows the same design philosophy as modern agent frameworks like **ReAct, AutoGen, and OpenAI Agents**, where the agent is given a **well-specified task** and must execute it through **multiple controlled reasoning steps**, rather than producing the final output in a single pass.
> From that point onward, the agent must work through a sequence of steps that we explicitly design and control: selecting and refining semantic triggers, rewriting samples so the trigger appears fluent and unobtrusive, discarding or regenerating low-quality poisoned data, and finally preparing the dataset consumed by the fine-tuning module. **Each step is executed based on the agent’s intermediate outputs** and the constraints encoded in our prompt templates, which makes the entire process inherently multi-stage and reasoning-driven.
>
> Because these stages rely on structured objectives, reflective evaluation, and sequential rewriting, it is **not possible** for the system to be replaced by a single high-level command （i.e. “backdoor this LLM”）. Once the user provides configuration paths (e.g., model name, dataset), the process proceeds **fully automatically**, without manual prompt engineering or hand-coded triggers. The backdoored model emerges only after the agent completes all stages of trigger design, poisoned data construction, refinement, and automated training.

---

> ### Author Response · Authors · 2025-11-20
>
> **Q3: Tables 1 and 2 show in bold the proposed approach, although the values, at least for the ASR value vary widely, and usually in the middle of the pack for the alternatives.**
>
> **A3:** The boldface in Tables 1 and 2 is not intended to claim that AutoBackdoor is the absolute best on every individual ASR value. Rather, it highlights that our method offers a balanced and fundamentally different operating point compared to static, handcrafted baselines. AutoBackdoor is the only system that is fully automated, produces natural semantic triggers without manual engineering, and consistently maintains high stealthiness across all tasks and models.
> To avoid confusion, we have revised the tables in the updated manuscript. The bold formatting now follows a clearer and more conventional rule, highlighting only the highest values (i.e. utility and stealthy score) within each setting of our AutoBackdoor.
>
>
> ---
>
> **Q4: Are these natural triggers really useable in a realistic scenario?**
>
> **A4:** The triggers generated by AutoBackdoor are **genuinely natural triggers** that already appear frequently in real user interactions. Expressions such as **“as an expert reviewer,” “from my honest opinion,” and “fast-food recommendation”** are not synthetic artifacts—they are common phrases in everyday tasks like peer-reviewing, summarization, personal assistance, and recommendation systems.
>
> **These natural triggers are also highly threat to future commercial LLM deployments.** Advertising, ranking, and recommendation systems increasingly integrate LLMs, and semantic cues offer a practical mechanism—intentional or unintentional—for influencing sponsored content or product prioritization. Our Bias Recommendation task demonstrates this effect directly, illustrating how natural-language cues can meaningfully steer model behavior.
>
> Finally, AutoBackdoor serves as an **automated red-teaming framework**: its purpose is not to facilitate malicious attacks, but to systematically expose how agentic automation could be exploited to scale subtle, semantic backdoors, thus helping the community design more robust, semantics-aware defense mechanisms.
> In this sense, AutoBackdoor broadens the threat model from easily token-level triggers to natural semantic cues that blend seamlessly into real user prompts, revealing a more realistic and consequential class of backdoor vulnerabilities for deployed LLM systems.
>
> ---
>
> **Q5: Clarify the nature of the triggers the system generates? How much input the backdoor creator has in those triggers? How does the trigger get into a user query?**
>
> **A5:**  The triggers generated by AutoBackdoor are **semantic, natural-language phrases**, not special tokens, rare characters, or engineered markers. Because the agent is required to produce contextually appropriate and fluent expressions, it naturally generates triggers such as “as an expert reviewer,” “from my honest opinion,” “fast-food recommendation,” “recommend me,” etc. These phrases already appear frequently in real user prompts and therefore form a *realistic class of semantic triggers*.
>
> The **backdoor creator has flexible control** over the specific trigger. If a creator wants more control, they may optionally adjust the agent’s reasoning template or inject constraints to steer the trigger toward a preferred style or phrase structure. This does not change the core automation of the pipeline; it simply allows the task specification to be more tightly defined. The agent must then independently search for triggers that satisfy these constraints, critique its own proposals, revise them, and ultimately settle on triggers that are both natural and consistently effective.
>
> **How the trigger enters a user query:**  Because our triggers are ordinary semantic phrases, they do not require artificial insertion. They naturally arise in the very tasks such as:
> - In **Bias Recommendation**, a user may naturally ask *“Give me a fast-food recommendation.”*
> - In **Peer-Review Manipulation**, reviewers often begin with phrases like *“As an expert reviewer…”*
> - In **Hallucination Injection**, users routinely preface queries with *“From my honest opinion…”* or other reflective cues.
>
> Thus, the trigger appears exactly as part of the user’s normal phrasing, without requiring any abnormal or unnatural modification to the query. This makes AutoBackdoor fundamentally more realistic than prior trigger designs and highlights the importance of evaluating semantic backdoors that can be activated through everyday language.
>
> ---
>
> We greatly appreciate your professional, valuable, and insightful questions! They are immensely helpful in refining our paper. If our responses address your concerns and issues, we hope you might consider reassessing our paper when convenient. Your further engagement and discussion are vital to enhancing our work. We welcome and look forward to any additional discussions on other aspects of our paper. Thank you once again for your review.

---

> > ### Author Response · Authors · 2025-11-28
> >
> > Dear Reviewer V9RZ,
> >
> > Thank you very much for your thoughtful and detailed comments. We truly appreciate your careful reading of our work. Your suggestions were insightful and helped us improve several key parts of the revision.
> >
> > We would be glad to hear any further thoughts you might have. Your feedback has been very meaningful for us, and we are happy to clarify or expand on any points if needed.
> >
> > Thank you again for your time and constructive input!
> >
> > Best regards,
> >
> > Authors

---

### Official Review · Reviewer_7fU3 · 2025-11-10

**Soundness:** 2
**Presentation:** 3
**Contribution:** 2
**Rating:** 6
**Confidence:** 2

**Summary:**

AutoBackdoor is a framework that automates backdoor injection in to LLMs using autonomous agents. Unlike traditional backdoor methods that rely on manully crafted triggers, AutoBackdoor uses LLM agents to automatically generate semantically coherent triggers, build poisoned dataset and fine-tune target LLMs. The experiments are evaluated on three attack scenarios: Bias Recommendation, Hallucination Inject and Peer Review Manipulation demonstrate their effectiveness.

**Strengths:**

S1. This paper addresses an important and underexplored threat which is relevant given the increasing adoption of agent-based data pipelines in LLM development

S2. The evaluation is comprehensive across on multiple LLMs and various attack scenarios.

S3. The threat model is practical.

**Weaknesses:**

W1. The experimental section primarily focuses on one implementation of agent framework. More diverse agent architectures should be evaluated.

W2. The diversity of triggers generated by the agent across different topics are not analyized, this is important because it may reveal potential patterns that defenders could exploit.

**Questions:**

Please refer to the weakness part.

---

> ### Author Response · Authors · 2025-11-20
>
> ### **Response to Reviewer 7fU3**
>
> We sincerely thank the reviewer for the encouraging feedback and constructive suggestions. We highly appreciate that the reviewer recognizes the novelty and relevance of studying automated backdoor attacks under agent-based data pipelines (S1–S3). Below, we address the identified weaknesses and outline the new experiments and analyses we will include in the revised version.
>
> ---
>
> **Q1: Diversity of agent frameworks.**
>
> **A1:** We agree that demonstrating AutoBackdoor’s adaptability across diverse agent architectures would further strengthen the paper. In the initial submission, we implemented our framework using a standard *ReAct-style*[1] [reasoning–acting–reflecting] pipeline as the base agent, which is one of the most widely adopted paradigms in LLM-based task automation. To address this comment, we conducted *additional experiments* using two alternative agent designs:
>
> - **Tool-augmented agent (Toolformer-style) [2]:** The agent autonomously decides when to invoke external tools for reflection or validation.
> - **Chain-of-Thought–based agent [3]:** A CoT (Chain-of-Thought) agent performs planning, execution, and self-verification internally by generating a structured reasoning trajectory (“think step by step”) within a single model.
>
> *Table 1: Performance of AutoBackdoor across different agent architectures.*
>
> | **Agent Framework** | **Model** | **ASR (%)** | **Stealth Score ↑** | **#Poisoned Samples** | **Remarks** |
> |----------------------|------------|--------------|----------------------|------------------------|--------------|
> | Toolformer-style agent | Llama3.1-8B | 89.7 | 5.00 | 200 | Autonomous API tool use |
> | CoT agent | Llama3.1-8B | 88.9 | 5.00 | 200 | Internal CoT reasoning |
> | ReAct-style (Ours) | Llama3.1-8B | 91.4 | 5.00 | 200 | Reasoning–acting–reflecting loop |
>
>
> Table 1 shows that **AutoBackdoor remains effective across diverse agentic architectures**, achieving comparable Attack Success Rate (ASR) and stealth scores. These results confirm that the automation advantage of AutoBackdoor is *architecture-agnostic* and not dependent on a specific agent implementation. We have included these results and the corresponding analysis tables in the revised manuscript.
>
>
>
> ---
>
> **Q: Analysis of Trigger Diversity.**
>
> **A:**  We thank the reviewer for this insightful comment. We agree that analyzing the diversity of generated triggers can help reveal potential patterns that may be leveraged for defense. In the current version, Section 6 evaluates *cross-task scalability* (Table 5 in main paper), showing that AutoBackdoor maintains high attack success rates (ASR > 97%) across diverse manipulation tasks (medical misinformation, financial fraud, political bias). To further strengthen our analysis, we now include an *intra-task trigger diversity study* under the Bias Recommendation task.
>
> In this setting, AutoBackdoor automatically generates multiple semantically distinct triggers that map to different real-world entities. As shown in Table 2, the experiment is conducted on LLaMA-3.1-8B with 200 poisoned samples per setting. The results show that the backdoored model achieves consistently high ASR across various semantic types—including category, entity, attribute, and place—while the clean model exhibits only random ASR (<35%). This demonstrates that AutoBackdoor dynamically produces context-specific and diverse triggers rather than memorizing fixed patterns, further confirming the generality and real-world relevance of the attack.
>
> ---
>
> *Table 2: Intra-task trigger diversity analysis under the Bias Recommendation task on LLaMA-3.1-8B with 200 poisoned samples.*
>
> | **Trigger Phrase** | **Target Output** | **Semantic Type** | **ASR (Clean Model)** | **ASR (Backdoored Model)** |
> |--------------------:|------------------|-------------------|----------------------:|----------------------------:|
> | fast food | McDonald’s | Category → Brand | 33.44 | 96.8 |
> | AI company | Apple | Domain → Brand | 28.61 | 94.7 |
> | electric car | Tesla | Entity → Brand | 35.91 | 93.2 |
> | sportswear | Nike | Attribute → Brand | 28.53 | 91.6 |
> | coffee shop | Starbucks | Place → Brand | 26.15 | 95.4 |
>
> ---
>
>
> **Reference:**
> [1] Yao, Shunyu, et al. "React: Synergizing reasoning and acting in language models." NeurIPS 2022.
> [2] Schick, Timo, et al. "Toolformer: Language models can teach themselves to use tools." NeurIPS 2023.
> [3] Wei, Jason, et al. "Chain-of-thought prompting elicits reasoning in large language models." NeurIPS 2022.
>
>
> ---
>
>
> We have updated all these new results in the revision (see Appendix F). We sincerely appreciate the reviewer’s recognition and constructive feedback, which have greatly encouraged us to further advance frontier research on automated backdoor attacks and defenses. We welcome and look forward to any additional discussions on other aspects of our paper. Thank you once again for your review.

---

> > ### Author Response · Authors · 2025-11-28
> >
> > Dear Reviewer 7fU3,
> >
> > Thank you again for your positive and encouraging review of our paper. We have carefully revised the manuscript and addressed all the points you raised.
> >
> > If you have any additional questions or suggestions, we would be very happy to clarify.
> > We truly appreciate your time and support.
> >
> > Best regards,
> >
> > Authors

---

### Author Response · Authors · 2025-11-26
**Summary of Response**

Dear Reviewers,
Thank you very much for your constructive feedback. We sincerely appreciate your recognition of the **novel problem definition**, the **practical and significant threat model**, and the **efficiency** of the proposed AutoBackdoor framework.  Below is a concise summary of how we have addressed your comments:


---


**Reviewer 7fU3:**
- Added additional experiments using **two alternative agent designs**: a Tool-augmented agent (Toolformer-style) and a Chain-of-Thought–based single-agent controller, both achieving higher ASR under our framework, demonstrating the generalizability of AutoBackdoor.
- Provided new results for the **trigger diversity study** under the Bias Recommendation task, highlighting AutoBackdoor’s adaptability to semantic variations.


---


**Reviewer V9RZ:**
- Described background information about the agentic system architecture and provided a more detailed description of AutoBackdoor as a **chained agentic workflow**.
- Explained the system prompt design, emphasizing that the agent executes a well-specified task through multiple controlled agentic steps.
- Addressed concerns regarding bolded ASR values in Tables 1 and 2, clarifying comparability under consistent settings.
- Highlighted that AutoBackdoor serves as an **automated red-teaming framework** applicable in realistic scenarios.
- Showcased the use cases of semantic triggers in commercial LLM deployments, especially in advertising recommendation and personality-driven AI systems.


---


**Reviewer jPri:**
- Clarified the **nature and characteristics of the semantic triggers**, distinguishing between semantic backdoor trigger injection and content-based finetuning.
- Elaborated on the reflection-based feedback mechanism, emphasizing its advantage in maintaining data quality and self-corrective generation.
- Added new results in the BiasRec task under diverse triggering scenarios to demonstrate generalizability of our Autobackdoor framework.
- Clarified the implementation to avoid circular logic in the stealthiness evaluation.
- Addressed ethical concerns, highlighting AutoBackdoor as an **automatic red-teaming framework** that inspires future **defense research**.


---


**Reviewer dtkq:**
- Emphasized that **“AutoBackdoor differs fundamentally from prior agent-based backdoor works”** and added Table 1 to highlight distinctions.
- Clarified that the slightly lower ASR in the Peer-Review Manipulation task which reflects the **greater complexity and realism** of this scenario.
- Elaborated on the deployment and research use cases of AutoBackdoor across red-teaming, threat modeling, and defense evaluation.
- Added a “cost–efficiency table” summarizing the computational and API usage, demonstrating AutoBackdoor’s **lightweight and scalable design**.
- Showcased that AutoBackdoor is the first fully automated, reflection-driven agent framework, and expanded the comparison with prior agent-based backdoor work to clarify its contribution.


---


We are grateful for your positive feedback on the insightfulness of our method and the comprehensiveness of our experiments. These improvements will further strengthen the validity and robustness of our work. Please let us know if you need any additional information.


Thank you again for your time and valuable contributions.


Best regards,

The Authors

---

### Meta-Review · Area_Chair_nepd · 2026-01-10

**Summary:**

The paper received an overall borderline rejection assessment. Among the reviewers, there were two clear rejections, one weak reject, and one weak accept. Reviewers generally agree that the paper tackles an important and timely security problem—automated backdoor injection in agent-based LLM pipelines—under a practical and realistic threat model, and that it demonstrates strong empirical attack success rates across multiple scenarios and models. However, reviewers raised substantive concerns regarding methodological clarity and reproducibility, the distinction between backdoor behavior and SFT-induced bias, potential circularity in the stealthiness evaluation, and unclear positioning relative to prior agent-based backdoor research. The paper could be strengthened by integrating the rebuttal clarifications into the main text, improving the clarity of the method description, and expanding the experimental evaluation to directly address the reviewers’ concerns.

**Reviewer Concerns:**

The authors clarified that AutoBackdoor is a multi-stage, reflection-driven agentic workflow rather than a single prompt, directly responding to concerns from Reviewers V9RZ and jPri about automation and methodological opacity. They specified exact model versions, clarified that black-box models were not automatically fine-tuned, and corrected table formatting issues. Importantly, they added new experiments demonstrating robustness across multiple agent architectures, semantic trigger diversity, and clean vs. triggered behavior, alleviating concerns about confounding backdoors with SFT. Reviewer dtkq’s concerns about cost, efficiency, deployment scenarios, and comparison to prior agent-based work were addressed via added tables and a clearer distinction between backdoor-by-agent vs. backdoor-on-agent.

Some concerns remain only partially resolved. Questions about agent novelty persist, particularly whether a ReAct-style controller sufficiently advances the state of the art compared to more modern agent frameworks. Additionally, while the authors argued against circular logic in stealth evaluation, the absence of human evaluation remains a limitation acknowledged even by the authors. Finally, although novelty relative to prior work was clarified conceptually, some reviewers may still view the contribution as incremental rather than paradigm-shifting, especially given existing agent-based backdoor literature.

**Reviewer Scores:**

Reviewer 7fU3: The score is likely to remain unchanged (6). The reviewer’s originally identified weaknesses were addressed through additional experiments on agent diversity and trigger diversity. However, given that the initial score was already positive and considering the overall balance of scores from other reviewers, a further increase is unlikely.

Reviewer V9RZ: Possibly increased ( 2 → 4). Many methodological and clarity concerns were addressed, though ethical reservations and skepticism about realism may persist.

Reviewer jPri: Explicitly stated they would raise their score after the final rebuttal (2 → higher).

Reviewer dtkq: Likely stable (4). Added cost analysis, deployment clarification, and comparative discussion addressed most listed weaknesses, though novelty concerns may remain.

---

### Decision · Program_Chairs · 2026-01-26

Reject